# The R251K Substitution in Viral Protein PB2 Increases Viral Replication and Pathogenicity of Eurasian Avian-like H1N1 Swine Influenza Viruses

**DOI:** 10.3390/v12010052

**Published:** 2020-01-02

**Authors:** Mengkai Cai, Ruting Zhong, Chenxiao Qin, Zhiqing Yu, Xiaoyan Wen, Junsi Xian, Yongjie Chen, Yu Cai, Heyou Yi, Lang Gong, Guihong Zhang

**Affiliations:** 1College of Veterinary Medicine, South China Agricultural University, Guangzhou 510462, China; caimengkai@126.com (M.C.); rutingzhong812@163.com (R.Z.); qinchenxiaol@163.com (C.Q.); zhiqingyu@stu.scau.edu.cn (Z.Y.); wenxiaoyanla@163.com (X.W.); xianjunsi123@163.com (J.X.); vetcyj@163.com (Y.C.); 18826271660@163.com (Y.C.); heyouyi@stu.scau.edu.cn (H.Y.); 2Key Laboratory of Zoonosis Prevention and Control of Guangdong Province, South China Agricultural University, Guangzhou 510462, China; 3National Engineering Research Center for Breeding Swine Industry, South China Agricultural University, Guangzhou 510462, China

**Keywords:** Eurasian avian-like H1N1 swine influenza viruses, *PB2* gene, PB2-R251K, pathogenicity, polymerase

## Abstract

The Eurasian avian-like swine (EA) H1N1 virus has affected the Chinese swine industry, and human infection cases have been reported occasionally. However, little is known about the pathogenic mechanism of EA H1N1 virus. In this study, we compared the mouse pathogenicity of A/swine/Guangdong/YJ4/2014 (YJ4) and A/swine/Guangdong/MS285/2017 (MS285) viruses, which had similar genotype to A/Hunan/42443/2015 (HuN-like). None of the mice inoculated with 10^6^ TCID_50_ of YJ4 survived at 7 days post infection, while the survival rate of the MS285 group was 100%. Therefore, a series of single fragment reassortants in MS285 background and two rescued wild-type viruses were generated by using the reverse genetics method, and the pathogenicity analysis revealed that the *PB2* gene contributed to the high virulence of YJ4 virus. Furthermore, there were 11 amino acid differences in PB2 between MS285 and YJ4 identified by sequence alignment, and 11 single amino acid mutant viruses were generated in the MS285 background. We found that the R251K mutation significantly increased the virulence of MS285 in mice, contributed to high polymerase activity and enhanced viral genome transcription and replication. These results indicate that PB2-R251K contributes to the virulence of the EA H1N1 virus and provide new insight into future molecular epidemiological surveillance strategies.

## 1. Introduction

Influenza A virus (IAV) is an important respiratory pathogen that continually impacts both the animal industry and human public health. The natural reservoir for IAV is thought to be wild waterfowl, but viruses frequently jump species barriers and infect humans and other mammals, including pigs, cats, and dogs [1]. Among these accidental hosts, swine has been recognized as one of the most important “mixing vessels” for the reassortment among avian and mammalian IAVs, because it displays both α-2,3 and α-2,6 receptors on their trachea cells. Those receptors are needed for human and avian influenza viruses infection respectively [2]. It has been thought that the 2009 pandemic influenza A H1N1 (pdm H1N1) virus was generated from co-infections by genetically distinct viruses in pigs [3].

The Eurasian avian-like swine (EA) H1N1 virus was first isolated from pigs in Northern Europe in 1979 [4], after which these viruses quickly spread out in Europe. Since 2005, EA H1N1 virus has been introduced into pigs in China and becomes the predominant virus [5]. Sporadic human infections caused by EA H1N1 virus have been recorded in European countries since 1986 [6,7,8,9,10]. In China, since the first human EA H1N1 virus infection in Jiangsu Province in 2011, five reports of infections with EA H1N1 virus in humans were reported [11,12,13,14,15]. A full-genome analysis revealed that these five human-isolated EA H1N1 viruses could be divided into two main genotypes, represented by A/Jiangsu/1/2011 (JS1-like) and A/Hunan/42443/2015 (HuN-like) [15,16]. In addition, EA H1N1 antibodies have been detected in swine farm residents and live poultry market workers [17], which has raised human public health concern that the EA H1N1 virus could cause a pandemic. Thus, a better understanding of the viral pathogenicity and public health risks of EA H1N1 virus is crucially needed, which can help develop effective control strategies and aid future pandemic preparedness.

There are many contributing factors of the virulence and host range to influenza A virus, and certain amino acid (aa) substitutions in these factors could alter the host range from avian to mammalian species as well as increase virulence in viral infection. For example, the receptor-binding specificity and virus transmission of pdm H1N1 virus is significantly affected by the amino acids at positions 222 and 226 in HA [18,19]. The L336M and T97I mutations in PA protein play important roles in polymerase activity and mouse pathogenicity of the pdm H1N1 virus and avian virus [20,21]. Additionally, regarding the highly virulent avian influenza viruses, the PB2 E627K mutation is the most well-characterized adaptation and virulence marker. This mutation has been found in the majority of H5N1 human infections and has demonstrated increased lethality in mice and has contributed to virus transmission in guinea pigs [22,23]. Apart from E627K described above, PB2 T271A mutation enhanced polymerase activity and virus growth of pdm H1N1 virus in mammalian hosts [24]. PB2 Q591R/K can compensate for the function of 627K and increase replication efficiency of pdm H1N1 virus in humans [25]. D701N [26] and E158G [27] in PB2 have similarly been identified to provide advantages to pathogenicity of pdm H1N1 virus in mice. However, compared with molecular pathogenic determinants known for avian and human influenza viruses, little has been known for swine influenza virus, especially for EA H1N1 virus. To investigate molecular pathogenic determinants of EA H1N1 virus, we compared the mouse pathogenicity of A/swine/Guangdong/YJ4/2014 (YJ4) [28] and A/swine/Guangdong/MS285/2017 (MS285) [29], belonging to the HuN-like virus, which is the main genotype causing human infections [16], and a new pathogenic determinant of EA H1N1 virus was identified by further analysis of a series of reassortant and mutant viruses.

## 2. Materials and Methods

### 2.1. Viruses and Cells

The two viruses A/swine/Guangdong/YJ4/2014 (YJ4 (H1N1)) and A/swine/Guangdong/MS285/2017 (MS285 (H1N1)) have been previously described (GISAID accession numbers EPI_ISL_249797 and EPI_ISL_293281, respectively) [28,29]. The viruses were amplified in Madin–Darby canine kidney cells (MDCK; ATCC, Manassas, VA, USA) at 37 °C, and the obtained viruses were collected and stored at −80 °C until use in experiments. MDCK cells, human lung adenocarcinoma epithelial cells (A549; ATCC, Manassas, VA, USA), and human embryonic kidney cells (293T; ATCC, Manassas, VA, USA) were propagated in Dulbecco’s modified essential medium (DMEM; Invitrogen, Carlsbad, CA, USA) containing 10% fetal bovine serum (FBS; Invitrogen, Carlsbad, CA, USA). All cells were cultured at 37 °C with 5% CO_2_.

### 2.2. Mouse Experiments

Six-week-old female BALB/c mice (Guangdong Medical Laboratory Animal Center, Guangdong, China) were anaesthetized with isoflurane and then inoculated intranasally with 10^6^ 50% tissue culture infectious dose (TCID_50_) of each virus in a total volume of 50 μL or mock inoculated with phosphate-buffered saline (PBS). After infection, the mice were monitored for 14 days for body weight and death rate, and mice with loss of more than 25% of their initial body weight were euthanized humanely. At 1, 3, and 5 days post-infection (dpi), three mice in each group were euthanized to determine virus replication in the nasal turbinate and lungs by TCID_50_ assay in MDCK cells.

### 2.3. Generation of Recombinant Viruses

The eight viral segments from YJ4 and MS285 viruses were cloned into the plasmid vector pHW2000 as previously described [30]. A series of single nucleotide mutation of the *PB2* gene in YJ4 viruses were introduced into the plasmids of MS285 virus using PCR-based site-directed mutagenesis with primers containing the desired mutations [31]. Primer sequences will be provided upon request. All plasmids were confirmed by Sanger sequencing. The eight plasmids (0.8μg each plasmid) were transfected into 293T cells in six-well plates (approximate 90% confluent) using 5 µL of Lipofectamine 3000 (Invitrogen, Carlsbad, CA, USA) in Opti-MEM (Invitrogen) containing 0.5 μg/mL tosylsulfonyl phenylalanyl chloromethyl ketone (TPCK)-treated trypsin (Sigma-Aldrich, Saint Louis, MO, USA). At 24 h post transfection, the supernatants containing rescued viruses were harvested and the viruses were propagated in MDCK cells. All propagated viruses were confirmed by RT-PCR and sequencing.

### 2.4. Growth Kinetics in Cells

The growth kinetics of recombinant viruses were compared in MDCK and A549 cells at a multiplicity of infection (MOI) of 0.01 TCID_50_/cell. After virus contact to the cells for 1 h, the cells were washed with PBS and further incubated in DMEM with 1 μg/mL TPCK-treated trypsin at 33 °C or 37 °C with 5% CO_2_. Culture supernatants were harvested at 12, 24, 36, 48, and 60 h post-inoculation (hpi) and stored at −80 °C. Viruses in the culture supernatants were titrated for infectivity by TCID_50_ assays using MDCK cells.

### 2.5. Viral Polymerase Activity Analysis

As previously described [32,33], 293T cells in 12-well plates (approximate 90% confluent) were co-transfected with pCAGGS plasmids encoding the *PB2* (wild-type or mutant), *PB1*, *PA*, and *NP* genes (300 ng each) of YJ4 and MS285 viruses, together with 100 ng of pPolI-NP-luc; and 10ng of pPL-TK (Promega, Madison, WI, USA) using 2.5 µL of Lipofectamine 3000 (Invitrogen, Carlsbad, CA, USA). Transfected cells were incubated at 33 °C or 37 °C. Cell extracts were harvested and then the luciferase activity was measured using a GloMax^®^ Discover System (Promega) according to the manufacturer’s instructions at 24 h post-transfection. All experiments were run in triplicate.

### 2.6. Quantitative RT-PCR Assays

The levels of viral NP segment viral RNA (vRNA), complement RNA (cRNA), and viral messenger RNA (mRNA) were measured by quantitative real-time PCR [34]. After infection with an MOI of 1 of indicated recombinant viruses in A549 cells for 24 h, total RNA was isolated using TRIzol reagent (Invitrogen, Carlsbad, CA, USA) according to the manufacturer’s instructions. Total RNA (1 μg) was reverse transcribed by MLV (Takara, Dalian, China) with either an oligo (dT) primer or a strand-specific primer. Strand-specific primer sequences were synthesized as follows (5′–3′): NP cRNA (GCTAGCTTCAGCTAGGCATCAGTAGAAACAAGGGTATTTTTCCTC), and NP vRNA (GGCCGTCATGGTGGCGAATAAATGGACGAAGGACAAGGGTTGC). Quantitative real-time PCR (qPCR) was performed with a GoTaq qPCR Master Mix (Promega) on the CFX96TM Real-Time System (Bio-Rad, Hercules, CA, USA). The following primers were used (5′–3′): 18S RNA (Forward: GTAACCCGTTGAACCCCATT, Reverse: CCATCCAATCGGTAGTAGCG) [35]; NP mRNA (Forward: CCAGATCGTTCGAGTCGT, Reverse: CGATCGTGCCTTCCTTTG); NP cRNA (Forward: GCTAGCTTCAGCTAGGCATC, Reverse: CGATCGTGCCTTCCTTTG); and NP vRNA (Forward: GGCCGTCATGGTGGCGAAT, Reverse: CTCAGAATGAGTGCTGACCGTGCC).

### 2.7. Western Blot Analysis

A549 cells were infected with an MOI of 1 of virus, with uninfected cells (mock) as a control. Cells were harvested 3, 6, 9, and 12 h post infection and lysed on ice in RIPA lysis buffer (Beyotime Biotechnology, Shanghai, China) containing protease inhibitor cocktails (Thermo Fisher, Waltham, MA, USA). The protein concentration in the lysates was assessed by using a bicinchoninic acid (BCA) protein assay kit (Beyotime, Shanghai, China). The samples were separated by SDS-PAGE and transferred to a nitrocellulose membrane (Millipore, Billerica, MA, USA). The membrane was blocked in 5% skim milk for 2 h, and then incubated overnight at 4 °C with anti-PB2 (GTX125926; Genetex, Irvine, CA, USA) or anti-NP (GTX125989; Genetex, Irvine, CA, USA) primary antibody. The membrane was washed three times with Tris-buffered saline containing Tween (TBST) for 5 min. Then, the membrane was incubated with goat anti-rabbit secondary antibody for 1 h in the dark at room temperature. The membrane was washed one time with TBST and then analyzed by using an Odyssey Infrared Imaging System (LI-COR Biosciences, Lincoln, NE, USA).

### 2.8. Statistics

Experimental results were analyzed by one-way ANOVA or Student’s *t*-test using SPSS software (Windows v21.0). The results are presented as the mean ± SD and a *p* < 0.05 was considered statistically significant.

### 2.9. Ethics Statement

All experimental animal work was carried out in accordance with animal ethics guidelines and approved protocols and was approved by the Experimental Animal Welfare Ethics Committee of the South China Agricultural University (No. 2019C011, approval date: June 4, 2019). All experiments with live viruses were conducted in a biosecurity level 2 laboratory.

## 3. Results

### 3.1. Pathogenicity of MS285 and YJ4 Viruses in Mice

The mouse model has been used commonly for evaluating the pathogenicity of influenza virus [36]. We previously showed that YJ4 virus is highly virulent in mice without adaptation [28]. To better understand the determinants of high virulence in mice infections, we chose MS285 virus, which has a similar genotype to YJ4 virus, and compared their virulence. Both YJ4 virus and MS285 virus belong to HuN-like virus, and are reassortants of EA H1N1, pdm H1N1, and classical swine H1N1 viruses (Table 1) [16]. Infection with 10^6^ TCID_50_ doses of MS285 or YJ4 viruses showed that all mice (*n* = 5) exhibited sustained body weight reductions (Figure 1A). The mean body weight of the mice in the MS285 group reached its lowest value at 8 dpi and returned to the approximate pre-infection weight by 14 dpi (Figure 1). In contrast, the mice in the YJ4 group lost weight more rapidly and died within six to seven days. In this experiment, YJ4 virus caused more severe body weight reductions and higher mortality than MS285 virus. These results suggest that YJ4 virus exhibited higher virulence than MS285 virus.

### 3.2. PB2 Substitution Increased the Pathogenicity of MS285 Virus in Mice

To evaluate which gene segments of YJ4 virus could enhance the virulence of the MS285 virus in the mouse model, we generated a series of reassortant viruses containing a single gene segment from YJ4 virus and the other seven segments from MS285 virus (Table 2). Groups of five mice each were inoculated intranasally with 10^6^ TCID_50_ of the reassortant viruses, and body mass and survival were observed daily for 14 days. Except for the r/MS285-PB2 virus, all other reassortant viruses did not cause mortality during the post-infection monitoring period, similar to the r/MS285 virus (Figure 2B). Conversely, the introduction of PB2 from YJ4 virus into MS285 virus enhanced the virulence of MS285 in mice. The mice infected with r/MS285-PB2 and r/YJ4 viruses showed marked body weight reductions after infection and died starting at 6 dpi (Figure 2). The mortality rate of r/MS285-PB2 and r/YJ4 group was 100%. In addition, overall body weight reductions in the group of r/MS285-PB1, r/MS285-HA, r/MS285-NP, and r/MS285-M were less pronounced than those in the r/MS285 group. And the introduction of PA, NA, and NS from the YJ4 virus into the MS285 virus produced no significant change in pathogenicity (Figure 2A). Since r/MS285 and r/MS285-PB2 viruses had the same backbone except for PB2, these data suggest that the *PB2* gene from the YJ4 virus plays an important role in the high virulence in mice.

### 3.3. The PB2 R251K Mutation Significantly Increased the Virulence of MS285 Virus in Mice

To identify the mutations responsible for the higher virulence, we compared the amino acid sequences of the *PB2* gene between MS285 and YJ4 viruses and generated recombinant viruses containing single amino acid mutations in MS285 PB2 (Table 3). As shown in Figure 3A, different effects on bodyweight loss were observed in mice infected with 10^6^ TCID_50_ of these reassortant viruses. Except for the r/MS285-251K virus, all other reassortant viruses were nonlethal in mice, similar to the r/MS285 virus (Figure 3B). Interestingly, the group of r/MS285-139V, r/MS285-375R, and r/ MS285-397A exhibited more significant weight loss than those in the r/MS285 group. The introduction of A221V, E249D, V255I, R340K, V451I, K493R, and I524T of PB2 into the MS285 virus produced no significant change in pathogenicity (Figure 3A). Notably, the R251K substitution in the *PB2* gene of these EA H1N1 viruses increased virus virulence in mice. All mice in the r/MS285-251K group exhibited drastic body weight reductions and all died within 7 dpi. Correspondingly, to determine whether change at position 251 in PB2 would alter the virulence of YJ4 virus in mice, we generated single amino-acid mutant virus in the YJ4 background and tested its virulence in mice. All infected mice died within five to seven days when infected with r/YJ4 virus, whereas three out of five mice survived when infected with r/YJ4-251R virus (60% survival rate; Appendix A). The other two survived mouse had mean maximum weight loss about 21.76% at 6 dpi and were seropositive with hemagglutination inhibition (HI) test at the end of experiment.

To further evaluate the contribution of the PB2-R251K mutation to replication in the respiratory organs of mice, mice were infected intranasally with 10^6^ TCID_50_ of r/MS285 and r/MS285-251K viruses, and nasal turbinate and lungs were collected at 1, 3, and 5 days post infection. As shown in Figure 3C,D, different levels of r/MS285 and r/MS285-251K viruses were detected in the nasal turbinate and lungs collected from infected mice. Compared with the r/MS285-251K group, which had a sustained increase in virus titers, the r/MS285 group demonstrated reductions in nasal turbinate and lung titers at 5 dpi. Of note, the virus titer in nasal turbinate of r/MS285-251K group was significantly higher than that of r/MS285 group at 5 dpi (*p* < 0.05, Figure 3C). In lung tissue, mice infected with r/MS285-251K also displayed higher viral titer than r/MS285 at 5 dpi (*p* < 0.05, Figure 3D). Overall, these results demonstrated that the PB2-R251K mutation increased the virulence of EA H1N1 virus in mice and correlated with increased viral replication efficiency in vivo.

### 3.4. PB2 R251K Mutation Enhanced the Replication Rate of MS285 in MDCK and A549 Cells

To examine whether the PB2-R251K mutation affects viral replicative capacity in vitro, we compared the growth properties of these two viruses in MDCK and A549 cells. As shown in Figure 4, the titers of r/MS285 and r/MS285-251K viruses progressively increased and peaked around 10^4^–10^8^ TCID_50_/mL at 48 hpi in MDCK and A549 cells. In MDCK cells, the r/MS285-251K virus replicated more efficiently than r/MS285 virus at 37 °C, and the titers of r/MS285-251K virus were more than 10-fold higher than those of the r/MS285 virus at 12, 24, and 36 hpi. At 33 °C, significantly higher titer of r/MS285-251K virus than r/MS285 virus was found in MDCK cells at 48 hpi (14.44-fold). In A549 cells, introduction of 251K to PB2 of MS285 virus significantly enhanced growth ability at 37 °C (36–48 hpi, about 16.63 to 19.63-fold), but this positive effect was not observed at 33 °C. Correspondingly, back substitution to 251R in the PB2 of YJ4 virus decreased the replication rate in A549 cells (48 hpi, 14.74-fold), but the differences are less dramatic in MDCK cells (Appendix A).

### 3.5. PB2-R251K Mutation Increased the Polymerase Activity and Genome Transcription of MS285

To understand whether the R251K substitution contributes to the polymerase activity of the EA H1N1 virus, we used mini-genome reporter assays in 293T cells at 33 °C and 37 °C, mimicking temperatures of the mammalian upper and lower respiratory tracts, respectively. Consistent with the result of growth kinetics at different temperatures, introduction of 251K into PB2 of MS285 virus significantly enhanced the polymerase activity at 37 °C (*p* < 0.001) with a 1.86-fold increase (Figure 5). There was no significant difference in polymerase activity at 33 °C. Since the PB2 protein participates in the transcription and replication of the viral genome, we then investigated whether R251K in PB2 influences the efficiency of virus genome transcription and replication. Consistent with the polymerase activity result, introducing R251K into the MS285 polymerase resulted in increased levels of viral NP segment cRNA and mRNA as demonstrated by specific viral mRNA, cRNA, and vRNA RT-qPCR assays (Figure 5B). For comparison, we also studied the accumulation of the viral PB2 and NP proteins of r/MS285 and r/MS285-251K viral isolates in A549 cells. Western blotting showed that the PB2 and NP proteins of r/MS285-251K virus was expressed at higher levels than those of r/MS285 virus. Overall, these data indicate that the introduction of 251K into PB2 of MS285 virus promotes viral polymerase activity and affects RNA synthesis and protein expression.

## 4. Discussion

In China, EA H1N1 virus first emerged in 2001 and has become predominant in swine since 2005 [5]. Subsequently, interaction of EA H1N1 virus with pdm H1N1 viruses generated multiple reassortants [37], one of which, HuN-like virus, has crossed the species barrier and caused three human infection cases [13,14,15]. In this study, both viruses, A/swine/Guangdong/YJ4/2014 and A/swine/Guangdong/MS285/2017, had a similar genotype to A/Hunan/42443/2015 (HuN-like), which was isolated from a child with severe pneumonia. By generating reassortant viruses, we found that the *PB2* gene is a major determinant of virulence in mice. Further analysis of the *PB2* gene suggested that the PB2-R251K mutation is associated with increased virulence in mice. In addition, virus titer studies in nasal turbinates and lungs showed that PB2-R251K is responsible for the increased viral replication efficiency in vivo.

Several virulence determinants in EA H1N1 viruses have been identified in previous studies. The HA-G225E mutation in EA H1N1 viruses affected receptor-binding preferences and increased the efficiency of virus assembly and budding, thereby increasing the transmission in guinea pigs [38]. The PB2-D701N mutation in EA H1N1 viruses contributed to viral replication and pathogenicity in mice [39]. Zhu W et al. reported that NP-Q357K of EA H1N1 viruses is pivotal for their enhanced pathogenicity phenotype in mice [16]. All of these amino acid changes affect viral replication in vitro. Similar to these previous findings, introduction of 251K into PB2 of EA H1N1 virus provided advantages in viral replication in MDCK and A549 cells. In addition, position 251 locates close to positions 253 and 256, which were demonstrated associated with the polymerase activity [40,41]. In our study, the polymerase complex containing the PB2-251K residue showed higher polymerase activity than that of wild-type influenza polymerase. These findings indicate that the PB2-R251K amino acid substitution increased the virulence of EA H1N1 viruses and could also be closely related to the enhancement of virus replication in vitro and the change in polymerase complex activity. To our knowledge, the present study is the first report identified that R251K mutation in the PB2 protein contributes to the virulence of EA H1N1 in vitro and in vivo.

Virulence of influenza A virus is affected by a great deal of factors. The mouse experiments of r/YJ4 and r/YJ4-251R viruses indicated that the single change at position 251 was insufficient to recover lethality of YJ4 virus up to 100% (Appendix A). Moreover, as shown in Figure 3A, mutants containing PB2-I139V, K375R, or T397A caused more significant weight loss, although they did not cause mortality in mice. Thus, the PB2-251K is not the only factor that determines the virulence in mice. The virulence of YJ4 virus may be affected by a joint effect of multiple amino acids and these factors, which may contribute to the high virulence of YJ4 virus remain to be further investigated.

Along with the PB1 and PA proteins, the PB2 protein is an important component of the polymerase complex of IAVs. The PB2 protein can bind to the 5′ cap of host pre-mRNAs and then participate in the transcription and replication of the viral genome [42]. Although the cap-binding domain has been studied extensively, its exact location remains controversial. Honda et al. demonstrated that the RNA cap-binding site is located approximately between residues 242–282 and 538–577 by UV-crosslinking studies [43]. However, mutagenesis experiments later confirmed that residues 363F and 404F played an essential role in cap binding [44]. It is possible that the 251 position is located in the cap-binding domain and thus affects the binding of host pre-mRNA to the PB2 protein. In our current study, we found that the introduction of 251K into PB2 resulted in the upregulation of RNA synthesis. In addition, we also found that this amino acid change conferred higher expression of the viral protein in human cells. In addition, we noted that a prior study identified two separate regions of PB2, 1–269, and 580–683, are capable of binding NP [45]. Iwai et al. determined that the N-terminal residues 1–256 participate in the binding process of PB2 and MAVS [46]. Coincidentally, the 251 position resides in these regions and may affect the interactions among PB2, NP, and MAVS. Additionally, arginine (R) and lysine (K) are both positively charged amino acids and have similar chemical characteristics, so the R-to-K mutations, like NP-R293K [47] and NA-R292K [48], have negligible impact on the basic structure and charge of the viral protein. Nevertheless, the replacement of arginine with the smaller lysine residue could affect the viral protein’s interaction with host factors and drugs, such as PA-R185K [49] and HA-R149K [50]. Similar effect may also pose by the R251K substitution in PB2 protein. However, these hypotheses remain to be further investigated.

In a recent study reported by Joseph et al. [51], the authors inferred that PB2-R251K played a critical role in the adaptation of avian influenza viruses into mammalian hosts by bioinformatics analysis. Of note, PB2 of the reassortant EA H1N1 virus (MS285 and YJ4 viruses) originates from the pdm/09 *PB2* gene segment, implying that introduction of PB2-251K in pdm H1N1 virus may enhance its replication and pathogenicity in humans. In addition, with continued circulation in swine populations, additional mutations are more likely to emerge, which in combination with the PB2-R251K mutation, may confer more efficient replication of the EA H1N1 virus in humans and lead to another human pandemic.

Taken together, the results of the present study demonstrated that the PB2-R251K mutation increased viral replication and pathogenicity in mice. This amino acid change conferred increased polymerase activity and more efficient replication in human cells. These findings not only provide new insights into the role of PB2-R251K on IAV pathogenicity but also suggest that PB2-R251K may have great significance for evaluating the health risks of IAVs.

## Figures and Tables

**Figure 1 viruses-12-00052-f001:**
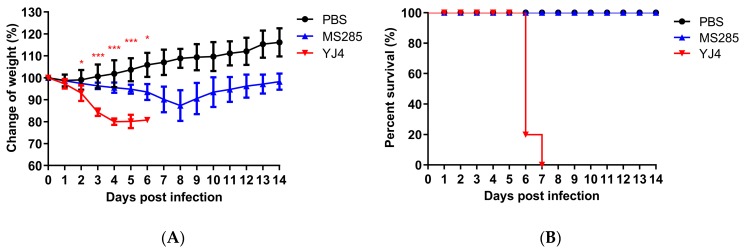
Weight changes and survival rates of mice infected with MS285 and YJ4 viruses. Six-week-old female BALB/c mice were inoculated intranasally with A/swine/Guangdong/YJ4/2014 and A/swine/Guangdong/MS285/2017 at 10^6^ TCID_50_. The body weight change rate (**A**) and survival rate (**B**) in the infected mouse were continuously recorded for 14 days. Mice with weight loss of more than 25% of their initial body weight were euthanized humanely. Significant body weight changes of the YJ4 viruses-inoculated mice compared with MS285 viruses-inoculated mice (* *p* < 0.05; *** *p* < 0.001).

**Figure 2 viruses-12-00052-f002:**
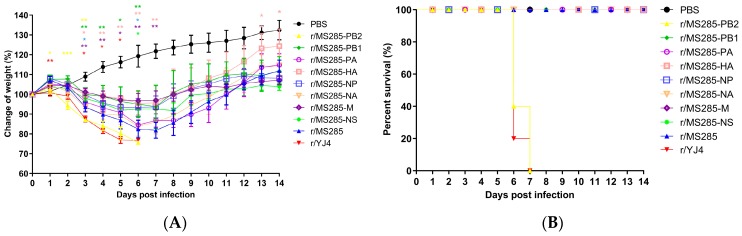
Weight changes and survival rates of mice infected with MS285 reassortant viruses. Six-week-old female BALB/c mice were inoculated intranasally with a series of reassortant viruses at 10^6^ TCID_50_. The body weight change rate (**A**) and survival rate (**B**) in the infected mice were continuously recorded for 14 days. Mice with weight loss of more than 25% of their initial body weight were euthanized humanely. Significant body weight changes of the single fragment recombinant virus-inoculated mice compared with r/MS285 virus-inoculated mice (* *p* < 0.05; ** *p* < 0.01; *** *p* < 0.001).

**Figure 3 viruses-12-00052-f003:**
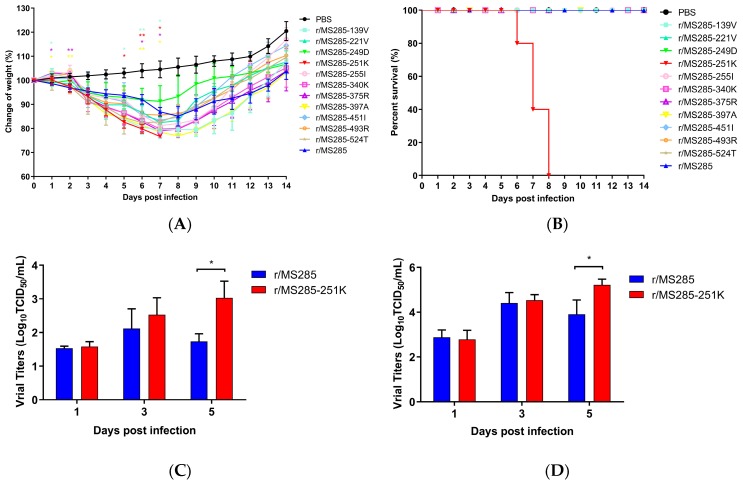
Weight changes, mortality, and replication of MS285 mutants in mice. Six-week-old female BALB/c mice were inoculated intranasally with 10^6^ TCID_50_ of reassortant viruses containing single amino acid mutations. The body weight change rate (**A**) and survival rate (**B**) in the infected mice were continuously recorded for 14 days. Mice with weight loss of more than 25% of their initial body weight were euthanized humanely. The average body weight change rates in each group are displayed with error bars representing the standard deviations (±). Significant body weight changes of the mutant virus-inoculated mice compared with r/MS285 virus-inoculated mice (* *p* < 0.05; ** *p* < 0.01). Replication efficiency of viruses in the nasal turbinate (**C**) and lungs (**D**) of infected mice. At 1, 3, and 5 days post-infection (dpi) with 10^6^ TCID_50_ of r/MS285 and r/MS285-251K viruses three mice were euthanized to collect the nasal turbinate and lungs. Virus replication was determined by TCID_50_ assay in MDCK cells.

**Figure 4 viruses-12-00052-f004:**
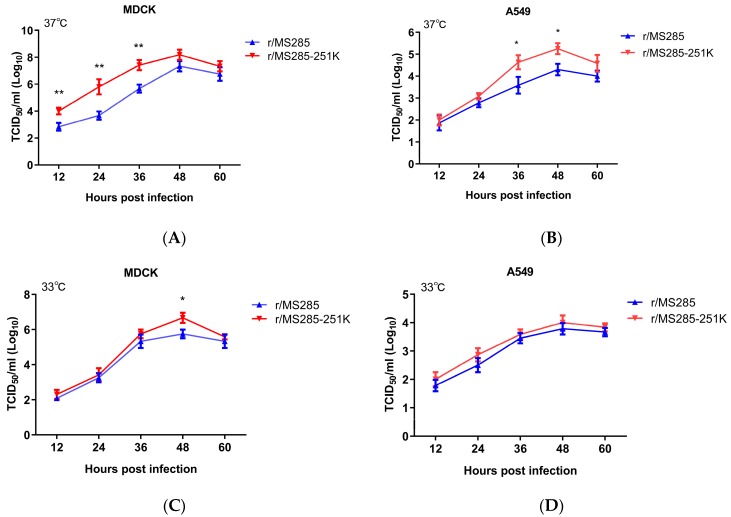
Growth kinetics of r/MS285 and r/MS285-251K viruses in MDCK and A549 cells. MDCK cells were infected at an multiplicity of infection (MOI) of 0.01 with r/MS285 and r/MS285-251K viruses and cultured at 37 °C (**A**) or 33 °C (**C**). A549 cells were infected at an MOI of 0.01 with r/MS285 and r/MS285-251K viruses and cultured at 37 °C (**B**) and 33 °C (**D**). Culture supernatants were harvested at 12, 24, 36, 48, and 60 hpi and subjected to TCID_50_ assay in MDCK cells. The results are expressed as the means ± SD (*n* = 3) and the statistical significance was calculated using one-way ANOVA. * *p* < 0.05; ** *p* < 0.01.

**Figure 5 viruses-12-00052-f005:**
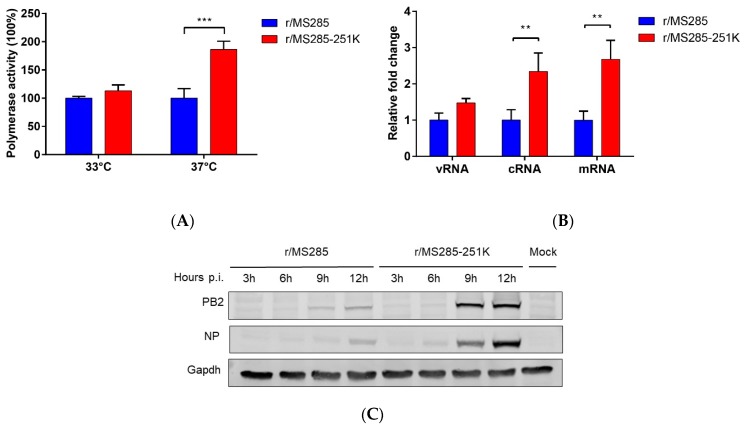
Effect of PB2-R251K on polymerase activity, RNA synthesis, and protein expression. (**A**) Polymerase activity was analyzed in 293T cells co-transfected with pPolI-NP-luc, pPL-TK, and expression plasmids encoding the *MS285 PB1, PA*, and *NP* genes and a PB2 clones (WT or R251K) at 33 °C or 37 °C. Relative polymerase luciferase activity compared to r/MS285 polymerase complex. (**B**) A549 cells were infected at an MOI of 1 with r/MS285 and r/MS285-251K viruses and cultured at 37 °C. After 24 hpi, the relative levels of viral NP segment viral RNA (vRNA), complement RNA (cRNA), and messenger RNA (mRNA) were quantified by strand-specific real-time RT-qPCR and normalized against 18S rRNA levels. The vRNA, cRNA, and mRNA values were expressed relative to the results for r/MS285. (**C**) A549 cells were infected at an MOI of 1 with r/MS285 and r/MS285-251K viruses and cultured at 37 °C. Cell lysates were prepared at 3, 6, 9, and 12 hpi to evaluate the expression levels of NP and PB2 by Western blotting. GAPDH was used as a loading control. In (**A**,**B**), the results are expressed as the means ± SD (*n* = 3), and the statistical significance was calculated using one-way ANOVA. ** *p* < 0.01, *** *p* < 0.001.

**Table 1 viruses-12-00052-t001:** Phylogenetic analyses of A/swine/Guangdong/YJ4/2014 and A/swine/Guangdong/MS285/2017.

Virus	Lineage Assigned to Gene Segment							
	PB2	PB1	PA	HA	NP	NA	M	NS
A/Jiangsu/1/2011 ^a^	EA	EA	EA	EA	EA	EA	EA	EA
A/Hebei-Yuhua/SWL1250/2012 ^a^	EA	EA	EA	EA	EA	EA	EA	EA
A/Hunan/42443/2015 ^a^	PDM	PDM	PDM	EA	PDM	EA	EA	CS
A/Fujian-cangshan/SWL624/2016 ^a^	PDM	PDM	PDM	EA	PDM	EA	PDM	CS
A/Tianjin-baodi/1606/2018 ^a^	PDM	PDM	PDM	EA	PDM	EA	PDM	CS
A/swine/Guangdong/YJ4/2014 ^b^	PDM	PDM	PDM	EA	PDM	EA	EA	CS
A/swine/Guangdong/MS285/2017 ^b^	PDM	PDM	PDM	EA	PDM	EA	EA	CS

Notes: CS, classical swine H1N1 lineage; EA, Eurasian avian-like swine (EA) H1N1 lineage; PDM, 2009 pandemic influenza A H1N1 lineage. ^a^ Eurasian avian-like swine H1N1 virus isolated from human [11,12,13,14,15]. ^b^ Eurasian avian-like swine H1N1 virus isolated from swine [28,29].

**Table 2 viruses-12-00052-t002:** Gene source of the single fragment reassortant influenza viruses.

Reassortant Virus	PB2	PB1	PA	HA	NP	NA	M	NS
r/MS285-PB2	YJ4	MS285	MS285	MS285	MS285	MS285	MS285	MS285
r/MS285-PB1	MS285	YJ4	MS285	MS285	MS285	MS285	MS285	MS285
r/MS285-PA	MS285	MS285	YJ4	MS285	MS285	MS285	MS285	MS285
r/MS285-HA	MS285	MS285	MS285	YJ4	MS285	MS285	MS285	MS285
r/MS285-NP	MS285	MS285	MS285	MS285	YJ4	MS285	MS285	MS285
r/MS285-NA	MS285	MS285	MS285	MS285	MS285	YJ4	MS285	MS285
r/MS285-M	MS285	MS285	MS285	MS285	MS285	MS285	YJ4	MS285
r/MS285-NS	MS285	MS285	MS285	MS285	MS285	MS285	MS285	YJ4
r/MS285	MS285	MS285	MS285	MS285	MS285	MS285	MS285	MS285
r/YJ4	YJ4	YJ4	YJ4	YJ4	YJ4	YJ4	YJ4	YJ4

**Table 3 viruses-12-00052-t003:** Amino acids differences in PB2 proteins among MS285 and YJ4 viruses.

Virus	139	221	249	251	255	340	375	397	451	493	524
A/swine/Guangdong/MS285/2017	I	A	E	R	V	R	K	T	V	K	I
A/swine/Guangdong/YJ4/2014	V	V	D	K	I	K	R	A	I	R	T

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
