# Peer review of "The R251K Substitution in Viral Protein PB2 Increases Viral Replication and Pathogenicity of Eurasian Avian-like H1N1 Swine Influenza Viruses"

_viruses, 2020, doi:10.3390/v12010052_

Round 1

Reviewer 1 Report

Cai et al. determined virulence and pathogenicity of two swine influenza viruses in mice and pinpointed a lethal determinant further using invitro settings such as virus growth and polymerase activity. Because the same genotype virus recorded human infection, it is important to identify pathogenic markers in relation to future surveillance of the HuN-like virus. Overall experiment is well designed and scientifically sound. For publication, there are a few points to be addressed in discussion. It is recommend that authors go through manuscript again with an English editing.

Abstract

Line 14

Please explain what the abbreviation (HuN-like) is. By only looking at the abstract, readers may be able to guess what HuN stands for, but it needs to be more specific.

Line 18

a series of reassortant viruses by substituting …. with …. and a recombinant MS285 virus were generated….

Introduction

Line 44

Phylogenetic analysis based on what a segment or genotype?

Line 51

… could alter the host range from avian to mammalian species as well as increase virulence in viral infection.

Line 60

Should be a bit more specific on what the advantage is.

Line 60-62

Please rephrase the sentence that figure out what was the main purpose of this study on either testing host range transition/expansion or finding pathogenic marker.

Method

Please inform sources of all the materials (animals, cells, and kits) used in this study such as product number, company, and country.

Provide a table or figure showing constitutions of viral segment and/or mutation used for viral growth curve and polymerase activity assay, defining its abbreviations.

Please disclose the biosafety level of facility where invitro and invivo studies were performed.

Line 79

Fifty percent tissue cell infectious dose (TCID50) ?

Line 86

Need a reference for the previous description.

Line 86

Single? or multiple? nucleotide mutations of the PB2 gene in …. virus were introduced to the plasmid of …. virus by using …..

Line 87

Add a reference of the mutagenesis PCR, or provide a protocol established in this study.

Line 96

After virus contact to … cells for 1 h, ….

Line 102-107

Please describe how the polymerase complex expresses the luciferase protein, or add a reference used for the test. Are the expressed NP-luc proteins intracellular or extracellular?

Line 109

Expend abbreviations, viral RNA, complement RNA, and viral messenger RNA.

Line 110

Total RNA was extracted from what experiment? Viral growth curve? Polymerase assay? If it was viral growth curve, at which time point were the cells harvested?

Line 123

Please describe the mock-infected cell.

Results

Line 152

How many mice were used for each group? Nowhere to find mice number but survival percentage.

Line 153-154

How significantly different? Which strain was the more virulent strain based on the mortality result?

Line 177

Supposed to mean the r/MS285-PB2 virus? If the recombinant YJ4 virus was used for comparison, please show it on figure 2.

How was the other reassortant viruses in body weight changes? Was there any significant differences between other groups?

Line 179

Since the r/MS285-PB2 and MS285 viruses had the same backbone except for PB2, …

Line 201

Three out of five mice inoculated survived when infected with r/YJ4-251R virus (Figure S1). The other two survived mice had/had not weight loss about 0000 and seropositive/seronegative with HI test at the end of experiment.

Line 209

At 5 dpi

Line 209-211

Please rephrase

Figure 3C

Please either rearrange or split the data by tissue factor

Line 228

Through the viral growth study (3.4), provide approximate number (average difference) when comparing two data points.

Line 230

As shown in Figure 4,

Line 232

The two viruses replicated high titer in 37 C about 00 (MDCK) and 00 (A549) TICD50/mL at 36 hpi

Line 252

introduction of 251K to PB2 of MS285 virus significantly…

Line 263

introduction of 251K to PB2 of MS285 virus promotes…

Figure 5A

Relative polymerase luciferase activity compared to r/MS285 polymerase complex.

Discussion

Could you explain or make a brief summary how other the genotypes shown in table 1 are different from HuN-like virus used in this study with respect to virulence and adaptation in mammalian or swine species? I see there was a previous study (Cao et al., 2019. Continuous evolution of influenza A viruses of swine from 2013 to 2015 in Guangdong, China)

Line 280-281

The EA H1N1 virus was introduced into Chinese pigs in 2005, and had reassortment with the pdm H1N1 virus in 0000. Consequently, the ….-like genotype became predominant in Chinese pigs and even had cross-species transmission to human in 0000-0000.

Line 284

What means similar genetic background? Genotype? Protein homology? If it was genotype, authors don’t have to call them the same genotype. If it was based on protein homology, add percentage of protein homology between viruses.

Line 304

Rephrase…

Line 307

… to recover lethality of YJ4 virus upto 100%

Line 315

Phenylalanine?

Line 328-331

Based on the virulence test in mice, the later isolate, MS285, is less virulent than the earlier isolate, YJ4. It seems that the HuN-like viruses got attenuated after four years of swine circulation. Although in short term the EA H1N1 lineage viruses acquired increased virulence in mice after reassortment with the pdm H1N1 viruses, this study also suggest that the virulence of the HuN-like genotype viruses decreased in mice. Authors should also discuss how and why they evolved to decrease virulence in terms of virus fitness and survival.

Author Response

Response to Reviewer 1 Comments

Point 1: Cai et al. determined virulence and pathogenicity of two swine influenza viruses in mice and pinpointed a lethal determinant further using invitro settings such as virus growth and polymerase activity. Because the same genotype virus recorded human infection, it is important to identify pathogenic markers in relation to future surveillance of the HuN-like virus. Overall experiment is well designed and scientifically sound. For publication, there are a few points to be addressed in discussion. It is recommend that authors go through manuscript again with an English editing.

Response 1: On behalf of all co-authors, I would like to take this opportunity to thank you for constructive suggestions and comments on our manuscript. We have revised the manuscript according to your comments and go through manuscript again with an English editing.

Point 2: Line 14. Please explain what the abbreviation (HuN-like) is. By only looking at the abstract, readers may be able to guess what HuN stands for, but it needs to be more specific.

Response 2: I have added the full name of "HuN-like" and it has been revised in Abstract.

Point 3: Line 18. a series of reassortant viruses by substituting …. with …. and a recombinant MS285 virus were generated….

Response 3: It has been revised accordingly in Abstract.

Point 4: Line 44. Phylogenetic analysis based on what a segment or genotype?

Response 4: Phylogenetic analysis based on a full-genome, it has been revised in the paragraph 2 in the Introduction.

Point 5: Line 51. … could alter the host range from avian to mammalian species as well as increase virulence in viral infection.

Response 5: It has been revised accordingly in the paragraph 3 in the Introduction.

Point 6: Line 60. Should be a bit more specific on what the advantage is.

Response 6: I have added the advantages in the paragraph 3 in the Introduction.

Point 7:  Line 60-62. Please rephrase the sentence that figure out what was the main purpose of this study on either testing host range transition/expansion or finding pathogenic marker.

Response 7: We rephrase the sentence as “However, compared with molecular pathogenic determinants known for avian and human influenza viruses, little has been known for swine influenza virus, especially for EA H1N1 virus. To investigate molecular pathogenic determinants of EA H1N1 virus, …” in this section.

Point 8:  Please inform sources of all the materials (animals, cells, and kits) used in this study such as product number, company, and country.

Response 8: The related content has been added in this section.

Point 9:  Provide a table or figure showing constitutions of viral segment and/or mutation used for viral growth curve and polymerase activity assay, defining its abbreviations.

Response 9: “Table 2 Gene source of the single fragment reassortant influenza viruses.” has been added in Section 3.2. Because of Table 3, showing the amino acids differences in PB2 proteins among MS285 and YJ4 viruses, we do not add another table to show the constitutions of recombinant viruses containing single amino acid mutation.

Point 10:  Please disclose the biosafety level of facility where invitro and invivo studies were performed.

Response 10: I have added the related content in Section 2.9 Ethics Statement

Point 10:  Line 79. Fifty percent tissue cell infectious dose (TCID50) ?

Response 11: I have added the full name of " TCID50" in this section.

Point 12:  Line 86. Need a reference for the previous description.

Response 12: I have added the reference in this section.

Point 13:  Line 86. Single? or multiple? nucleotide mutations of the PB2 gene in …. virus were introduced to the plasmid of …. virus by using …..

Response 13: It has been revised accordingly in Section 2.3.

Point 14:  Line 87. Add a reference of the mutagenesis PCR, or provide a protocol established in this study.

Response 14: I have added the reference in this section.

Point 15:  Line 96. After virus contact to … cells for 1 h, ….

Response 15: It has been revised accordingly in Section 2.4.

Point 16:  Line 102-107. Please describe how the polymerase complex expresses the luciferase protein, or add a reference used for the test. Are the expressed NP-luc proteins intracellular or extracellular?

Response 16: I have added the references in Section 2.5. And the expressed NP-luc proteins are intracellular

Point 17:  Line 109. Expend abbreviations, viral RNA, complement RNA, and viral messenger RNA.

Response 17: I have added the full names in Section 2.6.

Point 18:  Line 110. Total RNA was extracted from what experiment? Viral growth curve? Polymerase assay? If it was viral growth curve, at which time point were the cells harvested?

Response 18: We inserted “After infection with an MOI of 1 of indicated recombinant viruses in A549 cells for 24h,” in Section 2.6.

Point 19:  Line 123. Please describe the mock-infected cell.

Response 19: We inserted “with uninfected cells (mock) as a control.” in Section 2.7.

Point 20:  Line 152. How many mice were used for each group? Nowhere to find mice number but survival percentage.

Response 20: 5 mice were used for each group; the related content has been added in Section 3.1.

Point 21:  Line 153-154. How significantly different? Which strain was the more virulent strain based on the mortality result?

Response 21: We perform statistical analysis of body weight and the P-value were marked on the Figure 1A. The sentence has been revised, as “In this experiment, YJ4 virus caused more severe body weight reductions and higher mortality than MS285 virus. These results suggest that YJ4 virus exhibited higher virulence than MS285 virus.” in Section 3.1.

Point 22:  Line 177. Supposed to mean the r/MS285-PB2 virus? If the recombinant YJ4 virus was used for comparison, please show it on figure 2.

Response 22: We have added the body weight change rate and survival rate of YJ4 virus in figure 2 and perform statistical analysis of body weight and the P-value were marked on the Figure 2A.

Point 23:  How was the other reassortant viruses in body weight changes? Was there any significant differences between other groups?

Response 23: We perform statistical analysis of body weight and the P-value were marked on the Figure 2A and analyze the differences between other groups in Section 3.2, as “In addition, overall body weight reductions in the group of r/MS285-PB1, r/MS285-HA, r/MS285-NP and r/MS285-M were less than those in the r/MS285 group. And the introduction of PA, NA and NS from YJ4 virus into MS285 virus produced no significant change in pathogenicity (Figure 2A).”.

Point 24:  Line 179. Since the r/MS285-PB2 and MS285 viruses had the same backbone except for PB2, …

Response 24: It has been revised accordingly in Section 3.2.

Point 25:  Line 201. Three out of five mice inoculated survived when infected with r/YJ4-251R virus (Figure S1). The other two survived mice had/had not weight loss about 0000 and seropositive/seronegative with HI test at the end of experiment.

Response 25: It has been revised accordingly in Section 3.3.

Point 26:  Line 209. At 5 dpi

Response 26: It has been revised accordingly in Section 3.3.

Point 27:  Line 209-211. Please rephrase. Figure 3C. Please either rearrange or split the data by tissue factor

Response 27: We have revised Line 209-211 in Section 3.3 and split the data by tissue factor in Figure 3 (C-D).

Point 28:  Line 228. Through the viral growth study (3.4), provide approximate number (average difference) when comparing two data points.

Response 28: I have added approximate number in Section 3.4. As “As shown in Figure 4, the titres of r/MS285 and r/MS285-251K viruses progressively increased and peaked around 104–108 TCID50/ml at 48 hpi in MDCK and A549 cells. In MDCK cells, the r/MS285-251K virus replicated more efficiently than r/MS285 virus at 37 °C, and the titres of r/MS285-251K virus were more than 10-fold higher than those of the r/MS285 virus at 12, 24 and 36 hpi. And at 33 °C, significantly higher titre of r/MS285-251K virus than r/MS285 virus was found in MDCK cells at 48 hpi (14.44-fold). In A549 cells, introduction of 251K to PB2 of MS285 virus significantly enhanced growth ability at 37 °C (36-48 hpi, about 16.63 to 19.63-fold), but this positive effect was not observed at 33 °C. Correspondingly, back substitution to 251R in the PB2 of YJ4 virus decreased the replication rate in A549 cells (48 hpi, 14.74-fold ), but the differences are less dramatic in MDCK cells (Figure S2).”

Point 29:  Line 230. As shown in Figure 4,

Response 29: It has been revised accordingly.

Point 30:  Line 232. The two viruses replicated high titer in 37 C about 00 (MDCK) and 00 (A549) TICD50/mL at 36 hpi

Response 30: It has been revised in Section 3.4.

Point 31:  Line 252. introduction of 251K to PB2 of MS285 virus significantly…

Response 31: It has been revised accordingly in Section 3.5.

Point 32:  Line 263. introduction of 251K to PB2 of MS285 virus promotes…

Response 32: It has been revised accordingly in Section 3.5.

Point 33:  Figure 5A. Relative polymerase luciferase activity compared to r/MS285 polymerase complex.

Response 33: It has been added accordingly in Figure 5A.

Point 34:  Could you explain or make a brief summary how other the genotypes shown in table 1 are different from HuN-like virus used in this study with respect to virulence and adaptation in mammalian or swine species? I see there was a previous study (Cao et al., 2019. Continuous evolution of influenza A viruses of swine from 2013 to 2015 in Guangdong, China)

Response 34: We have added the related content in the paragraph 1 in the Discussion. “Since internal genes derived from different origins, these reassortant EA H1N1 viruses always showed different pathogenicity. Prior research reported that EA H1N1 viruses with M gene of EA origin exhibited higher virulence and replication than those with M gene of pdm/09 origin in mouse”

Point 35:  Line 280-281. The EA H1N1 virus was introduced into Chinese pigs in 2005, and had reassortment with the pdm H1N1 virus in 0000. Consequently, the ….-like genotype became predominant in Chinese pigs and even had cross-species transmission to human in 0000-0000.

Response 35: It has been added accordingly in the paragraph 1 in the Discussion.

Point 36:  Line 284. What means similar genetic background? Genotype? Protein homology? If it was genotype, authors don’t have to call them the same genotype. If it was based on protein homology, add percentage of protein homology between viruses.

Response 36: It was similar genotype. We have revised it in the paragraph 1 in the Discussion.

Point 37:  Line 304. Rephrase…

Response 37: We have revised as “Virulence of influenza A virus is affected by a great deal of factors.” in the paragraph 3 in the Discussion

Point 38:  Line 307. … to recover lethality of YJ4 virus upto 100%

Response 38: It has been added accordingly in the paragraph 3 in the Discussion.

Point 39:  Line 315. Phenylalanine?

Response 39: Yes, and we have revised it to the same format as elsewhere in the manuscript.

Point 40:  Line 328-331. Based on the virulence test in mice, the later isolate, MS285, is less virulent than the earlier isolate, YJ4. It seems that the HuN-like viruses got attenuated after four years of swine circulation. Although in short term the EA H1N1 lineage viruses acquired increased virulence in mice after reassortment with the pdm H1N1 viruses, this study also suggest that the virulence of the HuN-like genotype viruses decreased in mice. Authors should also discuss how and why they evolved to decrease virulence in terms of virus fitness and survival.

Response 40: Almost all avian influenza viruses contained 251R in PB2 protein and PB2-R251K mutation played a critical role in the adaptation of avian influenza viruses into mammalian hosts (Joseph et al., 2018 Adaptive evolution during the establishment of European avian-like H1N1 influenza A virus in swine). Although MS285 virus was isolated later than YJ4 virus and MS285 virus was less virulent than YJ4 virus, PB2-251K (YJ4 contained) has been present in swine since its appearance (e.g. A/swine/Saskatchewan/SD0221/2017 and A/swine/Oklahoma/A01785593/2018). Thus, we do not discuss the evolution and virulence between MS285 and YJ4 viruses.

Reviewer 2 Report

Major comments:

The authors had evaluated the growth kinetics of recombinant viruses in MDCK and A549 cells infected with influenza viruses at multiplicity of infection (MOI) of 0.1 or 1 TCID50/cell. These MOI seem to be very high, making difficult to evaluate the fitness of influenza viruses upon multiple cycles. Therefore, it is strongly suggested to perform the same experiments using lower MOI such as 0.01 or 0.001 per cell. Figure 1A: The graph strongly suggest that mice infected with MS285 influenza virus showed less weight lost that those infected with YJ4 influenza virus. Nonetheless, the authors should perform statistical analysis to corroborate this info. Figure 2A: It seems that some results were underexploited by the authors. Indeed, some recombinant MS285 viruses are more prone to induce higher weight loss than other ones. For instance, the weight lost observed in mice inoculated with M285-M virus seems to be higher than that observed in mice inoculated with M285-HA. Therefore, it is suggested to perform statistical analysis to evaluate if this assumption is true. Mostly, it would be very interesting to evaluate the total weight lost observed for each mice. This means, to evaluate the difference between the initial weight and the lowest weight measured for each animal. This approach could avoid some drawbacks due to the fact that the animals usually don't lose weight all at the same time after the infection The authors should discuss further how a single amino acid mutation in the PB2 protein can drastically affect the virulence of the genetically modified swine flu virus. This finding is very interesting, mainly because arginine and lysine are positively charged amino acids. Therefore, it deserves to be better discussed. In addition, the authors should emphasize in the discussion section of their manuscript that different results may have been obtained if they construct influenza viruses that carry more than one amino acid substitution in the PB2 protein (ie double substitutions, triple substitutions, and so on).

Minor comments:

Do HuN-like means Hunan like influenza viruses? If so, please include this information in text. Page 2, lane 56: Please, rewrite the following sentence: “Additionally, regarding the highly virulent avian influenza viruses, the PB2 E627K mutation is the most well-characterized adaptation and virulence marker.” instead “Additionally, the PB2 E627K mutation is the most well-characterized adaptation and virulence marker. Regarding the reverse genetics as well as Viral Polymerase Activity Analysis experiments, the authors should inform how many cells were seed into each cavity of the cell culture plaques. Page 6, line194: Please modify this sentence to ... overall body weight reductions in the r/MS285-249D group were less pronounced than... Page 8, lines 290-291: The sentence: “To our knowledge, our study is the first to reveal the effects of the R251K mutation in the PB2 protein on pathogenicity” seems to be incomplete.

Author Response

Response to Reviewer 2 Comments

Point 1:  The authors had evaluated the growth kinetics of recombinant viruses in MDCK and A549 cells infected with influenza viruses at multiplicity of infection (MOI) of 0.1 or 1 TCID50/cell. These MOI seem to be very high, making difficult to evaluate the fitness of influenza viruses upon multiple cycles. Therefore, it is strongly suggested to perform the same experiments using lower MOI such as 0.01 or 0.001 per cell.

Response 1: On behalf of all co-authors, I would like to take this opportunity to thank you for constructive suggestions and comments on our manuscript. We have performed the same experiments at an MOI of 0.01, and have revised the result and Figure 4. in Section 3.4.

Point 2:  Figure 1A: The graph strongly suggest that mice infected with MS285 influenza virus showed less weight lost that those infected with YJ4 influenza virus. Nonetheless, the authors should perform statistical analysis to corroborate this info.

Response 2: We perform statistical analysis of body weight and the P-value were marked on the Figure 1A.

Point 3:  Figure 2A: It seems that some results were underexploited by the authors. Indeed, some recombinant MS285 viruses are more prone to induce higher weight loss than other ones. For instance, the weight lost observed in mice inoculated with M285-M virus seems to be higher than that observed in mice inoculated with M285-HA. Therefore, it is suggested to perform statistical analysis to evaluate if this assumption is true. Mostly, it would be very interesting to evaluate the total weight lost observed for each mice. This means, to evaluate the difference between the initial weight and the lowest weight measured for each animal. This approach could avoid some drawbacks due to the fact that the animals usually don't lose weight all at the same time after the infection

Response 3: We perform statistical analysis of body weight and the P-value were marked on the Figure 2A and analyze the differences between other groups in this section. We inserted “In addition, overall body weight reductions in the group of r/MS285-PB1, r/MS285-HA, r/MS285-NP and r/MS285-M were less than those in the r/MS285 group. And the introduction of PA, NA and NS from YJ4 virus into MS285 virus produced no significant change in pathogenicity (Figure 2A).” in Section 3.2.

Point 4:  The authors should discuss further how a single amino acid mutation in the PB2 protein can drastically affect the virulence of the genetically modified swine flu virus. This finding is very interesting, mainly because arginine and lysine are positively charged amino acids. Therefore, it deserves to be better discussed.

Response 4: We inserted “Additionally, arginine (R) and lysine (K) are both positively charged amino acids and have similar chemical characteristics, so the R-to-K mutations, like NP-R293K and NA-R292K, have negligible impact on the basic structure and charge of the viral protein. Nevertheless, the replacement of arginine with the smaller lysine residue could affect the viral protein's interaction with host factors and drugs, such as PA-R185K and HA-R149K. Similar effect may also pose by the R251K substitution in PB2 protein.” in the paragraph 4 in the Discussion.

Point 5:  In addition, the authors should emphasize in the discussion section of their manuscript that different results may have been obtained if they construct influenza viruses that carry more than one amino acid substitution in the PB2 protein (ie double substitutions, triple substitutions, and so on).

Response 5: We inserted “Moreover, as shown in Figure 3A, mutants containing PB2-I139V, K375R or T397A caused more significant weight loss, although they did not cause mortality in mice. Thus, the PB2-251K is not the only factor that determines the virulence in mice. The virulence of YJ4 virus may be affect by a joint effect of multiple amino acids and these factors remains to be further investigated.” in the paragraph 3 in the Discussion.

Point 6:  Do HuN-like means Hunan like influenza viruses? If so, please include this information in text.

Response 6: I have added the full name of "HuN-like" in Abstract and Introduction.

Point 7:  Page 2, lane 56: Please, rewrite the following sentence: “Additionally, regarding the highly virulent avian influenza viruses, the PB2 E627K mutation is the most well-characterized adaptation and virulence marker.” instead “Additionally, the PB2 E627K mutation is the most well-characterized adaptation and virulence marker.

Response 7: It has been revised accordingly in the paragraph 3 in Introduction.

Point 8:  Regarding the reverse genetics as well as Viral Polymerase Activity Analysis experiments, the authors should inform how many cells were seed into each cavity of the cell culture plaques.

Response 8: It has been revised accordingly in Section 2.3 and 2.5.

Point 9:  Page 6, line194: Please modify this sentence to ... overall body weight reductions in the r/MS285-249D group were less pronounced than...

Response 9: Because the description of this sentence does not match the statistical analysis of body weight (Figure 3A), we deleted it and revised as " Interestingly, the group of r/MS285-139V, r/MS285-375R and r/MS285-397A exhibited more significant weight loss than those in the r/MS285 group."

Point 10:  Page 8, lines 290-291: The sentence: “To our knowledge, our study is the first to reveal the effects of the R251K mutation in the PB2 protein on pathogenicity” seems to be incomplete.

Response 10: We have moved the sentence from paragraph 1 to the paragraph 2 in the Discussion, and have revised as “To our knowledge, the present study is the first report identified that R251K mutation in the PB2 protein contributes to the virulence of EA H1N1 in vitro and in vivo.”

Reviewer 3 Report

In this article, the authors characterized two swine influenza A viruses (YJ4 and MS285) in a mouse model and found that YJ4 virus is highly virulent compared to MS285. By analyzing recombinant viruses, they identified that PB2 is the gene responsible for the high pathogenicity in mice. Further analysis of mutant viruses identified that PB2 R251K is a key mutation responsible for the virulence. The experimental approach is straightforward and convincing data are presented showing that PB2 residue 251 affects viral polymerase activity and pathogenicity in mice. However, many PB2 residues have already been identified to affect virus growth in mammalian hosts. The authors should appropriately acknowledge previous studies related to these PB2 mutations in Introduction or Discussion. 

Abstract Line 14: "HuN" should be spelled out for the first appearance.

Abstract Line 16-17: Mouse survival rate depends on the doses used for the study. The dose of the virus should be indicated in the sentence. 

Figure 1 legend: The authors need to state the number of mice used for the experiments.

Line 230: Figure 3 should be Figure 4.

Discussion Line 290-291: "To our knowledge, our study is the first to reveal the effects of the R251K mutation in the PB2 protein on pathogenicity" - the authors should acknowledge that PB2 residue 253 and 256, which are located close to the residue 251 have been reported to enhance polymerase activity and pathogenicigy in mammalian cells (J.Virol. 85:9641-5, 2011, and J.Virol. 83:1572-8, 2009). 

Author Response

Response to Reviewer 3 Comments

Point 1:  In this article, the authors characterized two swine influenza A viruses (YJ4 and MS285) in a mouse model and found that YJ4 virus is highly virulent compared to MS285. By analyzing recombinant viruses, they identified that PB2 is the gene responsible for the high pathogenicity in mice. Further analysis of mutant viruses identified that PB2 R251K is a key mutation responsible for the virulence. The experimental approach is straightforward and convincing data are presented showing that PB2 residue 251 affects viral polymerase activity and pathogenicity in mice. However, many PB2 residues have already been identified to affect virus growth in mammalian hosts. The authors should appropriately acknowledge previous studies related to these PB2 mutations in Introduction or Discussion.

Response 1: On behalf of all co-authors, I would like to take this opportunity to thank you for constructive suggestions and comments on our manuscript. I have added the related content in the paragraph 3 in the Introduction and in the paragraph 2 in the Discussion, as “Apart from E627K described above, PB2 T271A mutation enhanced polymerase activity and virus growth of pdm H1N1 virus in mammalian hosts. PB2 Q591R/K can compensate for the function of 627K and increase replication efficiency of pdm H1N1 virus in humans. D701N and E158G in PB2 have similarly been identified to provide advantages to pathogenicity of pdm H1N1 virus in mice.” and “In addition, position 251 locates close to positions 253 and 256, which were demonstrated associated with the polymerase activity.”

Point 2:  Abstract Line 14: "HuN" should be spelled out for the first appearance.

Response 2: I have added the full name of "HuN-like" in Abstract and Introduction.

Point 3:  Abstract Line 16-17: Mouse survival rate depends on the doses used for the study. The dose of the virus should be indicated in the sentence.

Response 3: I have added the dose of the virus in Abstract

Point 4:  Figure 1 legend: The authors need to state the number of mice used for the experiments.

Response 4: I have added the number of mice in Section 3.1.

Point 5:  Line 230: Figure 3 should be Figure 4.

Response 5: It has been revised accordingly.

Point 6:  Discussion Line 290-291: "To our knowledge, our study is the first to reveal the effects of the R251K mutation in the PB2 protein on pathogenicity" - the authors should acknowledge that PB2 residue 253 and 256, which are located close to the residue 251 have been reported to enhance polymerase activity and pathogenicigy in mammalian cells (J.Virol. 85:9641-5, 2011, and J.Virol. 83:1572-8, 2009).

Response 6: I have added the related content in the paragraph 2 in the Discussion, as “In addition, position 251 locates close to positions 253 and 256, which were demonstrated associated with the polymerase activity”

Round 2

Reviewer 2 Report

The current version of the manuscript brought significant improvements. However, some minor modifications should be made so that the manuscript is ready to be accepted.

Page 1, lines 35 – 37:  Among these accidental hosts, swine has been recognized as one of the most important “mixing vessels” for the reassortment among avian and mammalian IAV, because it displays both α-2,3 and α-2,6 receptors on their trachea cells  . Those receptors are needed for human and avian influenza viruses infencion respectively”  instead of  “ Among these accidental hosts, swine has been recognized as one of the most important “mixing vessels” for the reassortment of avian and mammalian IAVs since swine  has the α-2,3 and α-2,6 receptors which respectively needed for human and avian influenza viruses 37 [2].

Page 3, line 153: How many microliters of lipofectamine were added for each microgram of plasmid? This info should be added to the manuscript.

Page 4, line 223: “Infection with 106 TCID50 of MS285 or YJ4  viruses showed that all mice...” instead of “Infection with 106 TCID50 doses of the viruses showed that all mice...”

Page 5, line 267   “...were less pronounced than... ” instead of “were less than...”  

Page 6, line 304: “All infected mice died...” instead of “All the mice died...”  

Page 9, lines 438-443. These sentences are poorly written and should be carefully revised: “The EA H1N1 virus was introduced into Chinese pigs in 2005[5], and had reassortment with the  pdm H1N1 virus in 2010[37]. Consequently, the HuN-like genotype became predominant in Chinese  pigs[29,38], and even had cross-species transmission to human in 2015-2018[13-15]. Since internal  genes derived from different origins, these reassortant EA H1N1 viruses always showed different pathogenicity[28,38]. Prior research reported that EA H1N1 viruses with M gene of EA origin exhibited higher virulence and replication than those with M gene of pdm/09 origin in mouse[28].

The results showing the growth kinetics of recombinant viruses in MDCK and A549 cells infected with the multiplicity of infection (MOI) of 0.1 or 1 TCID50/cell could be shown in supplementary figures.

Author Response

Thank you for constructive comments on our manuscript. Followings are responses point by point to your comments.

Point 1:  Page 1, lines 35 – 37:  Among these accidental hosts, swine has been recognized as one of the most important “mixing vessels” for the reassortment among avian and mammalian IAV, because it displays both α-2,3 and α-2,6 receptors on their trachea cells  . Those receptors are needed for human and avian influenza viruses infencion respectively”  instead of  “ Among these accidental hosts, swine has been recognized as one of the most important “mixing vessels” for the reassortment of avian and mammalian IAVs since swine  has the α-2,3 and α-2,6 receptors which respectively needed for human and avian influenza viruses 37 [2].

Response 1: It has been revised accordingly.

Point 2:  Page 3, line 153: How many microliters of lipofectamine were added for each microgram of plasmid? This info should be added to the manuscript.

Response 2: The related content has been added in this section.

Point 3:  Page 4, line 223: “Infection with 106 TCID50 of MS285 or YJ4 viruses showed that all mice...” instead of “Infection with 106 TCID50 doses of the viruses showed that all mice...”

Response 3: It has been revised accordingly.

Point 4:  Page 5, line 267   “...were less pronounced than... ” instead of “were less than...”

Response 4: It has been revised accordingly.

Point 5:  Page 6, line 304: “All infected mice died...” instead of “All the mice died...” 

Response 5: It has been revised accordingly.

Point 6:  Page 9, lines 438-443. These sentences are poorly written and should be carefully revised: “The EA H1N1 virus was introduced into Chinese pigs in 2005[5], and had reassortment with the  pdm H1N1 virus in 2010[37]. Consequently, the HuN-like genotype became predominant in Chinese  pigs[29,38], and even had cross-species transmission to human in 2015-2018[13-15]. Since internal  genes derived from different origins, these reassortant EA H1N1 viruses always showed different pathogenicity[28,38]. Prior research reported that EA H1N1 viruses with M gene of EA origin exhibited higher virulence and replication than those with M gene of pdm/09 origin in mouse[28].

Response 6: We rephrase the sentences as “In China, EA H1N1 virus first emerged in 2001 and has become predominant in swine since 2005. Subsequently, interaction of EA H1N1 virus with pdm H1N1 viruses generated multiple reassortants, one of which, HuN-like virus, has crossed the species barrier and caused 3 human infection cases.” After consideration, we feel that “Since internal genes derived from different origins, these reassortant EA H1N1 viruses always showed different pathogenicity. Prior research reported that EA H1N1 viruses with M gene of EA origin exhibited higher virulence and replication than those with M gene of pdm/09 origin in mouse.” is not very relevant to this manuscript, so we delete these sentences.

Point 7:  The results showing the growth kinetics of recombinant viruses in MDCK and A549 cells infected with the multiplicity of infection (MOI) of 0.1 or 1 TCID50/cell could be shown in supplementary figures

Response 7: Thank you for constructive suggestions. We have reworked the experiment at an MOI of 0.01 based on your suggestions before. In this process, we reviewed some articles and found that the use of low MOI is more. And the growth kinetics of recombinant viruses in MDCK and A549 cells infected with the multiplicity of infection (MOI) of 0.1 or 1 TCID50/cell are similar to those of MOI of 0.01, so we do not add this part to supplementary figures.